# Classification Maps: A New Mathematical Tool Supporting the Diagnosis of Age-Related Macular Degeneration

**DOI:** 10.3390/jpm13071074

**Published:** 2023-06-29

**Authors:** Piotr Wąż, Katarzyna Zorena, Anna Murawska, Dorota Bielińska-Wąż

**Affiliations:** 1Department of Nuclear Medicine, Medical University of Gdańsk, 80-210 Gdańsk, Poland; 2Department of Immunobiology and Environment Microbiology, Medical University of Gdańsk, 80-210 Gdańsk, Poland; 3Department of Radiological Informatics and Statistics, Medical University of Gdańsk, 80-210 Gdańsk, Poland

**Keywords:** eye disease, age-related macular degeneration, mathematical modeling, molecular similarity, descriptors

## Abstract

Objective: A new diagnostic graphical tool—classification maps—supporting the detection of Age-Related Macular Degeneration (AMD) has been constructed. Methods: The classification maps are constructed using the ordinal regression model. In the ordinal regression model, the ordinal variable (the dependent variable) is the degree of the advancement of AMD. The other variables, such as CRT (Central Retinal Thickness), GCC (Ganglion Cell Complex), MPOD (Macular Pigment Optical Density), ETDRS (Early Treatment Diabetic Retinopathy Study), Snellen and Age have also been used in the analysis and are represented on the axes of the maps. Results: Here, 132 eyes were examined and classified to the AMD advancement level according to the four-point Age-Related Eye Disease Scale (AREDS): AREDS 1, AREDS 2, AREDS 3 and AREDS 4. These data were used for the creation of two-dimensional classification maps for each of the four stages of AMD. Conclusions: The maps allow us to perform the classification of the patient’s eyes to particular stages of AMD. The pairs of the variables represented on the axes of the maps can be treated as diagnostic identifiers necessary for the classification to particular stages of AMD.

## 1. Introduction

Age-Related Macular Degeneration (AMD) is the leading cause of central vision loss. It mainly affects people over 60 years of age and is responsible for about 50% of cases of blindness in the legal sense [1,2]. It is a chronic, progressive disease of the outer layers of the central part of the retina and the choroid. The number of people suffering from AMD is expected to increase. This is related to the extension of life of the human population and increasing exposure to risk factors for degenerative changes in the macula [1,3].

In the United States, 8 million Americans suffer from early AMD, and more than 1 million will develop advanced AMD in the next 5 years [2]. It is predicted that by 2050, 1 in 10 people over 50 years old in the US will be diagnosed with AMD [3]. It is estimated, that approximately 1.5 million people suffer from macular degeneration in Poland, of which 130,000 are patients with the more dangerous, exudative form of AMD [4]. Worldwide, the predicted number of people with AMD will increase to 288 million in 2040 (196 million in 2020) [5]. Due to the highest incidence of AMD in Europe, it is expected that the number of cases in this region will, in the future, be second only to Asia [6]. The frequency of occurrence of AMD in Europe is 12.33%, in Asia it is 7.38% and in Africa it is 7.53% [5,6]. This frequency increases with age—the percentage of patients is 13.4% for people over 60, and in the group of people aged 40–59, only 2.8%. The frequency of occurrence of advanced AMD at age 70 is 1.4%, rising to 5.6% at age 80 and 20% at age 90.9 [7].

The development of a breakthrough research method, which is ophthalmoscopy, i.e., the examination of the fundus of the eye, is considered to be the beginning of the science of retinology. It became possible thanks to Herman von Helmholtz, who, in 1851, built an ophthalmoscope. The macula was first mentioned at the end of the 18th century, when Samuel Thomas Soemmerring discovered and described in detail the yellowish area in the posterior pole of the retina. The first to describe symmetrical changes in the fundus of the eye were two researchers in 1875—Jonathan Hutchinson and Warren Tay. However, it was only 10 years later that Otto Haab gave these changes the name of a separate disease entity—senile macular degeneration [8]. Initially, the progress of knowledge about this disease was slow and concerned risk factors and the first classifications of the disease. In 1967, J. Donald Gass explained the pathogenesis of central vision loss and described the stages of AMD development [9,10].

Only in the last 20 years has there been a dynamic development of knowledge about AMD.

In particular, Optical Coherence Tomography (OCT) [11,12] allowed visualization of retinal structures in vivo. It is a diagnostic tool that uses low-coherence reflectometry to obtain images of living tissues. Its current spectral generation (high-resolution OCT) has expanded the knowledge of the morphological features present in AMD in the retina, RPE and choroid, thus enabling the further identification of new structural markers of the disease, such as ellipsoidal zone disorder, hyperreflective foci and drusenoid subretinal deposits. It is a non-invasive, non-contact method of retinal imaging [13]. The time delay and intensity of light scattered or reflected from tissues for tomographic imaging of their internal structure are measured. This is achieved by scanning tissues with high resolution, higher than in magnetic resonance imaging, computed tomography, or medical ultrasonography. Although OCT is used, e.g., in cardiology, oncology and gastroenterology, it has found the greatest application in ophthalmology due to the structure of the eye and its relationship with light [14,15,16]. It is used to image the structures of the anterior and posterior segment of the eyeball. It enables visualization of retinal structures and its morphology, and the obtained images can be analyzed using algorithms that calculate the thickness of the retina. The OCT test is also used to monitor the disease over time, as well as to assess the effectiveness of treatment [17]. A new imaging technology, OCT Angiography (OCTA), has recently gained popularity and is increasingly used in routine ophthalmic practice. OCT angiography creates structural images of blood flow in the retinal and choroid plexuses by combining the properties of the well-known OCT with the function of motion contrast. It can visualize vessels in different layers of the retina and choroid, as well as pathological neovascularization. This method allows for a quantitative assessment of blood flow in the retina and the optic disc [18].

Recently, many articles on the etiology, pathophysiology or treatment of the exudative form of AMD have been published [19,20,21,22]. For example, Tadao Maeda et al. discussed trends of stem cell therapies in AMD [23]. Jianhang Yin et al. studied safeguarding genome integrity during gene-editing therapy in a mouse model of age-related macular degeneration [24]. Wonyoung Jung et al. analyzed a relation between AMD and the risk of end-stage renal disease by using nationwide, population-based cohort data in Korea [25]. María Sanabria et al. studied the impact of COVID-19 on the quality of life of patients with AMD [26]. Yeong Choi et al. investigated the roles of miRNAs in the aqueous humor of patients with typical AMD and polypoidal choroidal vasculopathy using next-generation sequencing and quantitative PCR [27].

The macula is a circular area in the posterior pole, located between the temporal vascular arcades, 5–6 mm in diameter. It is responsible for the central field of vision in the area of 15–20${}^{\circ }$, precise close vision, color recognition and a sense of contrast. The inner layers of the macula contain a yellow pigment—xanthophyll and carotenoids—lutein and zeaxanthin in a higher concentration than in the peripheral retina. The fovea is a depression in the inner layer of the retina in the center of the macula with a diameter of 1.5 mm [28,29].

The Retinal Pigment Epithelium (RPE) is made up of a single layer of hexagonal cells containing numerous melanosomes in the apical pigment layer. At the base, RPE cells connect to the deeper layer of the retina with Bruch’s membrane, while their appendages extend between the photoreceptor layer of the neurosensory retina. In the posterior pole, in the foveal projection, RPE cells are taller and thinner with more melanosomes. The function of RPE cells is to create an external blood–retinal barrier thus preventing the passage of fluid from the choriocapillaries to the subretinal space. RPE cells take part in the active transport of water and ions from the subretinal space; the transport of nutrients and the removal of metabolic products; the storage, metabolism and transport of vitamin A; the facilitation of the metabolic transformation of photoreceptors; maintaining an optimal environment for the retina; and the RPE dye absorbs the scattered light falling on the retina [28,29].

Bruch’s membrane separates the RPE layer from the choriocapillaries and participates in the removal of waste products outside the retina. Its integrity with the RPE is disturbed by, among others, the presence of drusen or neovascular membrane (Choroidal Neovascularization—CNV) present in the wet form of the disease [28].

The pathogenesis of AMD is not fully understood. The development of the disease is influenced by oxidative stress, atherosclerosis-like changes, RPE cell dysfunction, various genetic variants, neovascularization as well as inflammation and altered immune responses of tissues [29]. The above processes disturbs the proper functioning of the central retina, including photoreceptors, RPE cells, Bruch’s membrane and choriocapillaries. Due to the area of the retina affected by the degenerative process, which is the macula, vision is impaired very quickly. Metabolic, functional, genetic and environmental factors overlap with the onset of the disease and its progression. One of the models for the development of advanced neovascular AMD suggests that the accumulation of drusen disrupts the connection between the RPE and the choroid, thereby causing hypoxia. Hypoxia induces the expression of VEGF-A (Vascular Endothelial Growth Factor—VEGF) and other proangiogenic factors for the formation of new, pathological vessels [30].

Increasing the number of pro-angiogenic factors, such as vascular endothelial growth factor and Platelet-Derived Growth Factor (PDGF), or decreasing the number of anti-angiogenic factors, such as Pigment Epithelium-Derived Factor (PEDF) and endostatin, may play a key role in the pathophysiology of the disease and, for this reason, are currently considered possible therapeutic targets [31,32].

Accumulation of drusen results from an abnormal influx of lipids to and from the RPE [33,34]. Drusen lipids come mainly from RPE and photoreceptors. The blood supply from the choroid provides a small amount of them, while the drusen proteins come from both the choroid vessels and the entire body [35]. The opposite is true for fats that build atherosclerotic plaques, in which lipids, proteins and lipoproteins come from peripheral blood [36].

The hemodynamic model of AMD pathogenesis takes into account the similarity between drusen and atherosclerotic plaques [37]. They have many common elements that support their common pathophysiology, e.g., Vitronectin (VTN), complement component 3 (C3), amyloid (beta, P), apolipoproteins, esterified and unesterified cholesterol, Matrix Metalloproteinases (MMP) and calcium [35,36].

Lipids deposited in the sclera increase its rigidity and choroidal vascular resistance, reducing choroidal blood flow and increasing vascular-capillary pressure, which leads to the formation of CNV. At the same time, the deposition of lipids in Bruch’s membrane causes the degeneration of elastin and collagen, as well as the formation of deposits of the basal lamina and drusen. Degeneration of elastin and collagen leads to calcification, increased VEGF-A concentration and, ultimately, the formation of CNV [37,38,39].

Currently, a greater share of chronic inflammation, endothelial dysfunction and oxidative stress in the pathogenesis of AMD is recognized in the deposition of lipid deposits [40]. Activation of the immune system in AMD patients causes pathological accumulation of lipids [41]. The accumulation of deposits under the neurosensory retina itself is a normal process of retinal aging. It becomes pathogenic only if the function of the complement system is disturbed (probably as a result of oxidative stress) and uncontrolled thus contributing to cell damage and apoptosis [36]. The pro-inflammatory environment of the retina, which is promoted by the RPE response to heterogeneous loads, seems to be a key modulator of CNV development and progression. Active factors of the complement system C3a and C5a are strong chemotactic substances that recruit leukocytes to the choroid and stimulate RPE cells to secrete pro-angiogenic VEGF [42]. The oxidative stress of the RPE caused by the products of photo-oxidation activates the complement components in the RPE. An autoimmune reaction induced by oxidative damage causes complement deposition in the retina [43,44].

The initial clinical symptoms of AMD are characterized by the presence of drusen, i.e., deposits of extracellular matrix and pigment, which most often form in the macula on the border of the choroid and the RPE. Based on the size and number of drusen, the presence of atrophy or neovascularization, AMD is divided into four stages of increasing severity [45,46]. Early and intermediate AMD is characterized by the presence of small or large drusen and RPE irregularities. Forms of late AMD include geographic atrophy and neovascularization, both of which can lead to severe central visual impairment and legal blindness due to degenerative and neovascular changes in the macula. Currently, neovascular AMD can be controlled with anti-angiogenic agents that block vascular endothelial growth factor. Despite this, most treated patients still suffer from visual impairment as they develop fibrosis and atrophy, and more than one-third of patients show long-term resistance or loss of response to the drug.

In this article, we propose a non-standard approach to deriving information about AMD. We demonstrate that a graphical representation of the considered data, constructed in this work (*classification maps*), is a new tool supporting the diagnosis of AMD.

Classification studies are a valuable source of information in various fields of science [47]. The classification problem is related to similarity studies [48]. The similarity between objects is not unique if they are described by several different characteristics [49]. The degree of similarity depends on the selected characteristics, the number of characteristics considered, and the mathematical measure that determines the relationship between different characteristics. In computational science we deal with characteristics expressed numerically. In the theory of molecular similarity, such characteristics are called “descriptors” [50,51]. The descriptors are applied in methods used in the theory of molecular similarity. The assumption “The molecules that are similar in certain aspects have similar properties” is a cornerstone of the methods of *Quantitative Structure-Activity/Property Relationships* (QSAR/QSPR) [52,53,54]. Such techniques are applied to predict the activity, reactivity or properties of new molecules.

It is not obvious how to represent the molecular structure by introducing the decsriptors and how to define the similarity measure. For example, we have used frequencies and intensities of molecular spectra to construct new kinds of molecular descriptors [55]. We have shown that the new descriptors correctly represent the molecular structure. An analysis of the infrared spectra of chloronaphthalenes illustrated this theory. As a graphical representation of these results, we have introduced classification maps. By analyzing the patterns of these maps, the similarities between the objects can be easily observed. Additionally, the correlations between the descriptors represented on the axes of the maps can be revealed [56].

An analogous methodology was applied to derive the information in other fields of science, i.e., in social science studying groups of individuals [57,58] or in bioinformatics characterizing the biological (DNA, RNA, protein) sequences [59,60]. Similarity studies of the sequences using graphical techniques are commonly used approaches. Each method reveals different aspects of similarity and still new approaches are being constructed in bioinformatics [61,62,63,64,65,66,67,68,69,70,71]. For example, we created *Spectral-Dynamic Representation of DNA Sequences* [65] or *4D-Dynamic Representation of DNA/RNA Sequences* and applied this bioinformatics method to studies on the origin of the SARS-CoV-2 virus [60].

In this work, we extend the applications of such an approach by constructing the classification maps for AMD. Variables such as CRT (Central Retinal Thickness) [72,73,74], GCC (Ganglion Cell Complex) thickness [75,76], MPOD (Macular Pigment Optical Density) [77,78], ETDRS (Early Treatment Diabetic Retinopathy Study), Snellen and Age have been used as the descriptors and are represented on the axes of the maps. There are numerous charts used for visual acuity testing, but the most common are Snellen and ETDRS charts. A detailed comparison between the two charts can be found in [79]. The classification maps represent a new graphical diagnostic aid for the detection of age-related macular degeneration.

## 2. Materials and Methods

A total of 132 eyes (66 patients) were examined and classified to the AMD advancement level based on the four-point Age-Related Eye Disease Scale (AREDS) [80]. AREDS is the most widely used classification among many classification systems of AMD. According to this scale, AMD can be classified as follows:*AREDS 1* group (as a control group)—no or only a few small drusen with a diameter of <63 μm;*AREDS 2* group—early form of AMD—the co-occurrence of numerous small drusen with a diameter of >15 μm, several drusen with a diameter of 63–125 μm or RPE abnormalities in the form of increased pigmentation or depigmentation;*AREDS 3* group—moderate AMD—numerous medium-sized drusen, at least one large druse > 125 μm in diameter, geographic atrophy not occupying the center of the macula;*AREDS 4* group—advanced form of AMD—geographic atrophy of the RPE with involvement of the macula, neovascular maculopathy, which includes: CNV, i.e., pathological vessels originating from the choroid, serous or hemorrhagic retinal detachment or RPE, exudation and hard, fibrovascular proliferations under the retina and under the RPE, discoid scar (choroidal fibrosis) [45,81].

The AREDS 1 group constituted 32 eyes without features of AMD. Due to the lack of changes in the eye fundus, patients from the AREDS 1 group constituted the control group. AREDS 2 constituted 37 eyes, and AREDS 3 constituted 33 eyes. The group with the most advanced form of AMD, AREDS 4, constituted 30 eyes.

It should be noted that in some cases, AMD changes were detected only in one eye or each eye the disease was of different AMD advancement level according to AREDS.

All patients underwent outpatient examinations in the years 2016–2017 at the UCK Ophthalmology Clinic in Gdańsk. The diagnosis of AMD was based on the current standards and recommendations of the Polish Society of Ophthalmology, in accordance with the guidelines of the American Academy of Ophthalmology (AAO) [82]. Data on the stage of the disease and the general health of the patients were obtained on the basis of: medical history, STARS form (Simplified Théa AMD Risk-assessment Scale), measurement of resting blood pressure and pulse and detailed ophthalmological examination, including SOCT macula (Spectral Optical Coherence Tomography) and macular pigment optical density. Each patient was informed about the essence of these tests.

The criteria for inclusion in the group of patients with AMD and in the group of patients without AMD features were age >55 years and age-related macular degeneration in the grades qualifying them according to the AREDS scale according to the guidelines of the Polish Society of Ophthalmology.

The calculations were performed using R programming language [83,84].

The median, minimum and maximum values were used to describe quantitative variables. Differences in the examined variable value distributions for the AMD advancement groups were analyzed using the non-parametric Kruskal–Wallis test. In the case of obtaining statistically significant results for this test, appropriate post-hoc tests were performed.

In this work, an ordinal regression is applied, implemented using a generalized linear model. The method can predict the value of an ordinal variable, a variable for which only the relative ordering between its different values is important. This method can also be regarded as a kind of classification. In the examined example, the ordinal variable (and simultaneously the dependent variable) is the degree of advancement of AMD. Quantities such as CRT, average GCC thickness, MPOD, ETDRS, Snellen and Age were used in the analysis. The measurements were performed using Zeiss Cirrus HD-OCT model 400 (Carl Zeiss Meditec, Inc., Dublin, CA, USA).

As a consequence, a tool indicating the degree of development of AMD was created.

Due to a small number of patients, the construction of multivariate ordinal regression models was limited to designing models containing two independent variables. The article presents only those pairs of independent variables for which the fitting coefficient vectors were statistically significant. For these pairs, Odds Ratios (OR) and Confidence Intervals (CI) were determined, as well as thresholds for individual values of ordinal variables. Based on the created theoretical models and calculated probability values, classification maps were created. The two-dimensional maps show the values of the independent variables represented on the axes of the maps. The model-predicted probability of classification to a specific value of the ordinal variable is marked on the map with a specific color (see subsequent section).

The assumed significance level was α = 0.05. (The significance level α is the predetermined acceptable risk of making an error of type I (the recognition of a true null hypothesis as false). This value is used to determine the threshold in the values of deviations above which the test selects an alternative hypothesis. This means that each *p*-value with the significance level lower than α allows us to conclude that the obtained result is statistically significant).

## 3. Results and Discussion

Clinical and ophthalmological characteristics of patients with AMD according to AREDS and the control group are summarized in Table 1.

Kruskal–Wallis test results for all variables shown in Table 1 are statistically significant.

The results of the post-hoc tests showed that for the CRT variable, differences in the distributions of the variable values occur between all groups. In the case of the GCC variable, a statistically significant result was discovered only between the control group and AREDS 3. For ETDRS, only in the case of the distributions of the variable values in the control group and AREDS 3, the result of the post-hoc test was not statistically significant. An identical result was obtained for the Snellen variable, i.e., there was no statistical significance between the distributions of values of this variable in the control group and AREDS 3. In the case of the MPOD variable, there were differences between the distributions of the variable values in the AREDS 4 and the other groups. Other relations between groups for this variable were not statistically significant. For the Age variable, only differences in the distributions of the variable values between the control group, AREDS 3, and AREDS 4 were statistically significant.

If the CRT value increases by 1 unit, the odds that the patient’s eye will have a higher stage of AMD is about 0.7% higher (Table 2). If the ETDRS value increases by 1 unit, the odds that the patient’s eye will be in a higher stage of AMD is about 6.4% lower (Table 2).

If the CRT value increases by 1 unit, the odds that the patient’s eye will have a higher stage of AMD is about 0.7% higher (Table 3). If the Snellen value increases by 0.1 units, the odds that the patient’s eye will be in a higher stage of AMD is about 23.5% lower (Table 3).

If the CRT value increases by 1 unit, the odds that the patient’s eye will have a higher stage of AMD is about 1.1% higher (Table 4). If the Age value increases by 1 unit, the odds that the patient’s eye will be in a higher stage of AMD is about 8.4% higher (Table 4).

If the GCC value increases by 1 unit, the odds that the patient’s eye will have a higher stage of AMD is about 8.7% lower (Table 5). If the ETDRS value increases by 1 unit, the odds that the patient’s eye will be in a higher stage of AMD is about 9.5% lower (Table 5).

The ordinal regression model for the GCC and Snellen values is analogous to the model for the GCC and ETDRS: if the GCC and Snellen values increase, the odds that the patient’s eye will have a higher stage of AMD decreases. If the GCC value increases by 1 unit, the odds that the patient’s eye will have a higher stage of AMD is about 7.8% lower (Table 6). If the Snellen value increases by 0.1 units, the odds that the patient’s eye will be in a higher stage of AMD is about 35.2% lower (Table 6).

If the GCC value increases by 1 unit, the odds that the patient’s eye will have a higher stage of AMD is about 6.4% lower (Table 7). If the Age value increases by 1 unit, the odds that the patient’s eye will be in a higher stage of AMD is about 12.6% higher (Table 7).

If the ETDRS value increases by 1 unit, the odds that the patient’s eye will have a higher stage of AMD is about 6.7% lower (Table 8). If the Age value increases by 1 unit, the odds that the patient’s eye will be in a higher stage of AMD is about 5.3% higher (Table 8).

Analogous to the previous cases, if the Snellen value increases by 0.1 units, the odds that the patient’s eye will have a higher stage of AMD is about 24% lower (Table 9). If the Age value increases by 1 unit, the odds that the patient’s eye will be in a higher stage of AMD is about 5.7% higher (Table 9).

The data presented in Table 2, Table 3, Table 4, Table 5, Table 6, Table 7, Table 8 and Table 9 were used to create classification maps (Figure 1, Figure 2, Figure 3, Figure 4, Figure 5, Figure 6, Figure 7 and Figure 8), in particular: the classification maps CRT–ETDRS (Figure 1) correspond to Table 2, CRT–Snellen maps (Figure 2) to Table 3, CRT–Age maps (Figure 3) to Table 4, GCC–ETDRS maps (Figure 4) to Table 5, GCC-Snellen maps (Figure 5) to Table 6, GCC–Age maps (Figure 6) to Table 7, ETDRS–Age maps (Figure 7) to Table 8, and Snellen–Age maps (Figure 8) correspond to Table 9.

Classification maps show the probability (greater than 50%) of the occurrence of the stages of AMD depending on the variables (“descriptors” in the theory of molecular similarity). Probability intervals are denoted by different colors in the figures. The pair of descriptors represented on the axes of the maps classify the patient’s eyes to particular stages of AMD (control group, AREDS 2, AREDS 3 or AREDS 4).

Figure 1, Figure 2 and Figure 3 show classification maps only for the control group and AREDS 4. This means that pairs of descriptors representing the map axes are good identifiers for both cases. The diagnosis of AREDS 4 can be made on the basis of descriptor pairs: (CRT, ETDRS), Figure 1; (CRT, Snellen), Figure 2; or (CRT, Age), Figure 3. Using these descriptor pairs, intermediate AMD stages (AREDS 2 and AREDS 3) cannot be diagnosed.

Patterns in Figure 1 and Figure 2 are similar, suggesting a correlation between ETDRS and Snellen.

The pairs (GCC, ETDRS), (GCC, Snellen) and (GCC, Age) can be used to diagnose all stages of AMD: Control group, AREDS 2, AREDS 3 and AREDS 4. The corresponding classification maps are shown in Figure 4, Figure 5 and Figure 6, respectively.

In Figure 7 and Figure 8, similarly to the maps shown in Figure 1, Figure 2 and Figure 3, the conditions for probabilities higher than 50% were met only in the cases of the control group and AREDS 4. This means that the pairs of variables, (ETDRS, Age) in Figure 7 and (Snellen, Age) in Figure 8, diagnose these two stages of AMD.

The quality of the model is very high if an area representing a probability greater than 90% appears on the map (purple color in the figures). The following stages of AMD can be diagnosed with high precision:AREDS 4 using (CRT, ETDRS) variables—Figure 1;AREDS 4 using (CRT, Snellen) variables—Figure 2;AREDS 4 using (CRT, Age) variables—Figure 3;AREDS 4 using (GCC, ETDRS) variables—Figure 4;AREDS 4 using (GCC, Snellen) variables—Figure 5;AREDS 4 using (GCC, Age) variables—Figure 6;AREDS 4 using (ETDRS, Age) variables—Figure 7.

Since the maps for which the probabilities were smaller than 50% are not shown, all other maps presented in this article can also be used to diagnose the stages of AMD with a good precision.

It should also be noted that large areas on the maps, representing the probabilities of the occurrence of stages of AMD, mean large ranges of values of variables that classify these stages. Diagnosis may be easier in the cases of the large areas (high precision of the variable value measurement is not required).

## 4. Conclusions

It should be noted that the given odds and their impact on the change of the dependent variable values are correct if the second variable is constant. The ranges of the values of all independent variables in the computational models are restricted to the collected data (Table 1).

The quality of all classification models considered in this work is approximately the same. However, the models containing the GCC variable better estimate the probability of classification to all four groups of AMD than the other models. Models not containing the GCC variable can indicate with a high probability the control group and the AREDS 4 group, only. Among these models, the one with a paired CRT and Age variables is better than the others.

Although the result of the Kruskal–Wallis test for MPOD variable grouped according to the AREDS scale was statistically significant, it was not possible to create a two-dimensional ordinal regression model containing the MPOD variable with statistically significant fitting coefficients.

Summarizing, the classifications maps constitute a new supporting diagnostic graphical tool for the detection of the Age-Related Macular Degeneration. The values of the variables presented on the axes of the maps classify the patient’s eyes to particular groups (control group, AREDS 2, AREDS 3 or AREDS 4) (Figure 9). Using this alternative computational approach, all stages of AMD can be diagnosed with high or good accuracy.

In the framework of the created model, we try to predict stages of AMD in cases with rather limited data. We are looking for the dependence of AMD stages on various variables. Due to the limited data, we could only create the two-dimensional models. We consider this method to be a pilot study for multidimensional models. The map shapes indicate correlations between variables—this suggests that a modification of norms and deriving some analytical mathematical formulas describing this issue can be feasible.

The presented method both diagnoses the AMD stage and predicts (on the basis of the collected data) a change of this stage as a function of the values of the variables used.

A new method is always an added value that allows us to verify more complex and precise methods. In our method, we use simple analytical functions in explicit form. This description facilitates the interpretation of the obtained results and creates a platform to understand more complicated processes.

The values of some variables have been collected using simple methods and simple and cheap devices. It is important that mathematical models could be created using these variables, while more advanced tools and methods are not available.

Personalized medicine is based on using specific features of individual patients in order to make optimized decisions about their treatment. Frequently, the differences between these features are small. Therefore, high precision mathematical models are a prerequisite for their potential usefulness in solving problems arising in the applications of personalized medicine. The computational approach presented in this work, in particular the classification maps, constitutes an example of such a model.

## Figures and Tables

**Figure 1 jpm-13-01074-f001:**
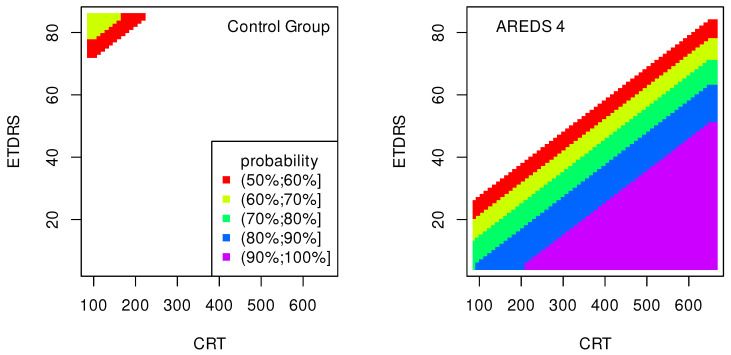
Classification maps CRT–ETDRS.

**Figure 2 jpm-13-01074-f002:**
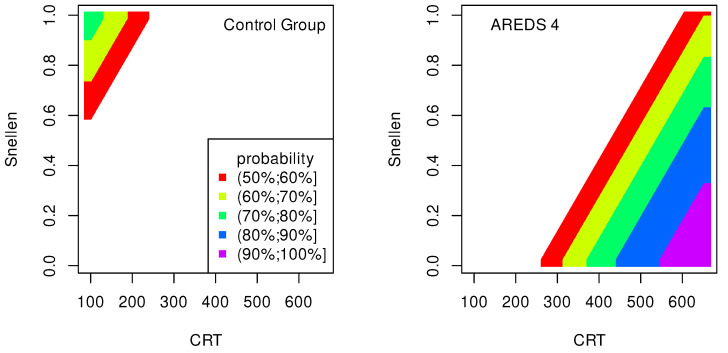
Classification maps CRT–Snellen.

**Figure 3 jpm-13-01074-f003:**
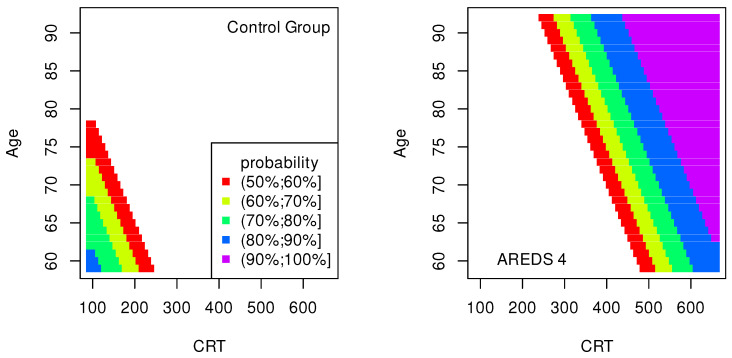
Classification maps CRT–Age.

**Figure 4 jpm-13-01074-f004:**
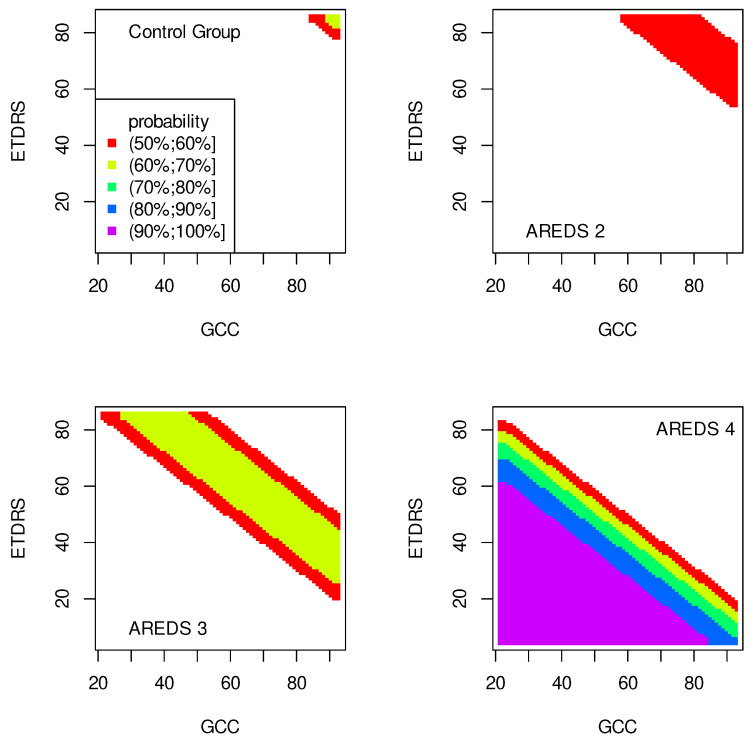
Classification maps GCC–EDTRS.

**Figure 5 jpm-13-01074-f005:**
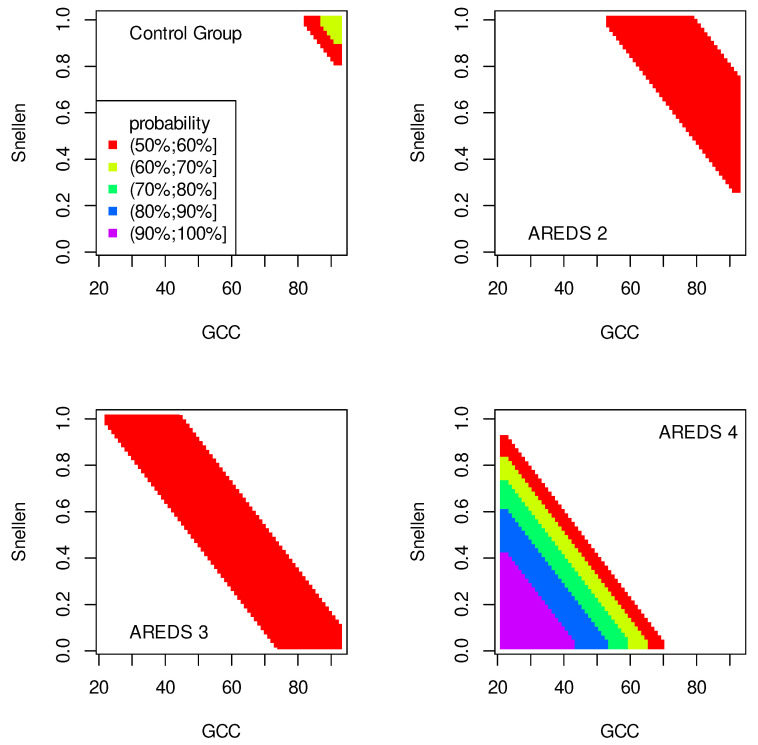
Classification maps GCC–Snellen.

**Figure 6 jpm-13-01074-f006:**
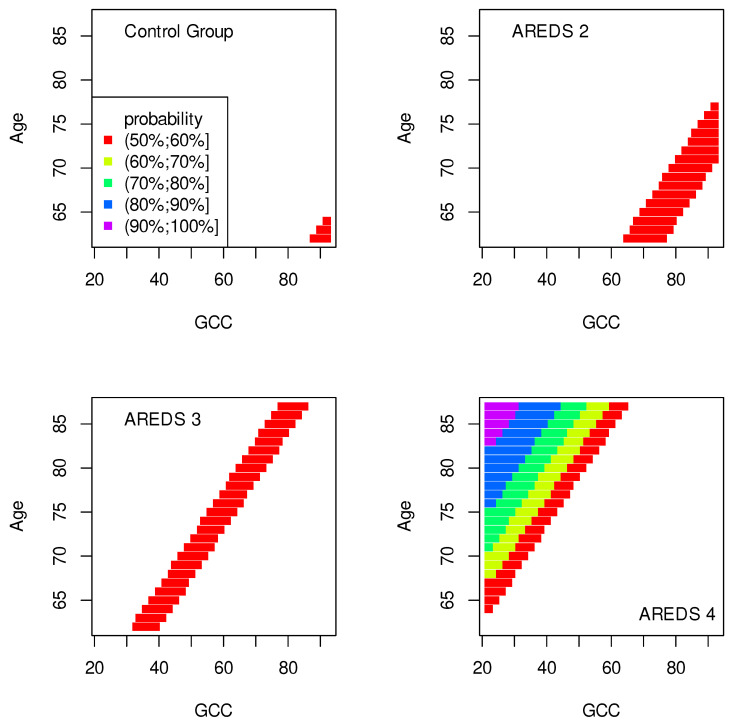
Classification maps GCC–Age.

**Figure 7 jpm-13-01074-f007:**
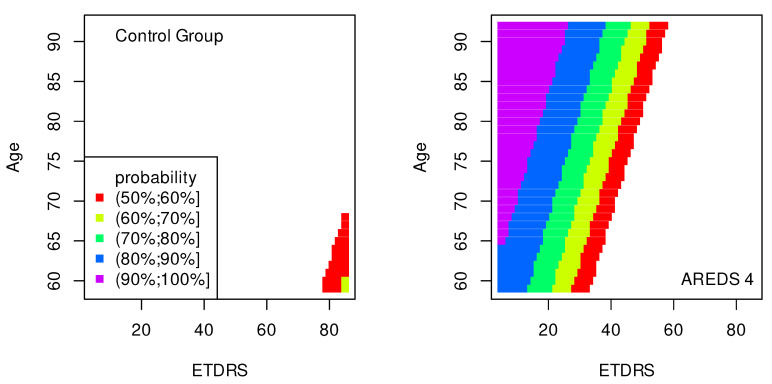
Classification maps EDTRS–Age.

**Figure 8 jpm-13-01074-f008:**
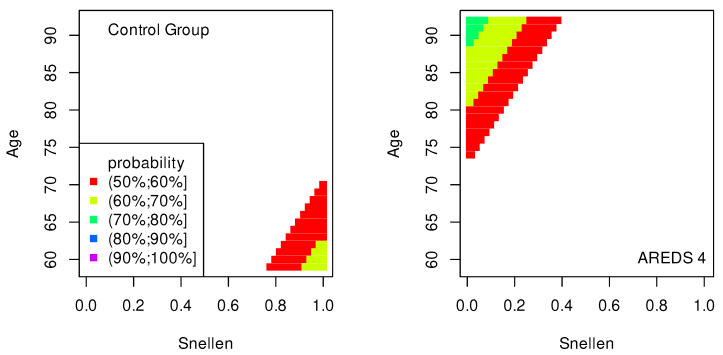
Classification maps Snellen–Age.

**Figure 9 jpm-13-01074-f009:**
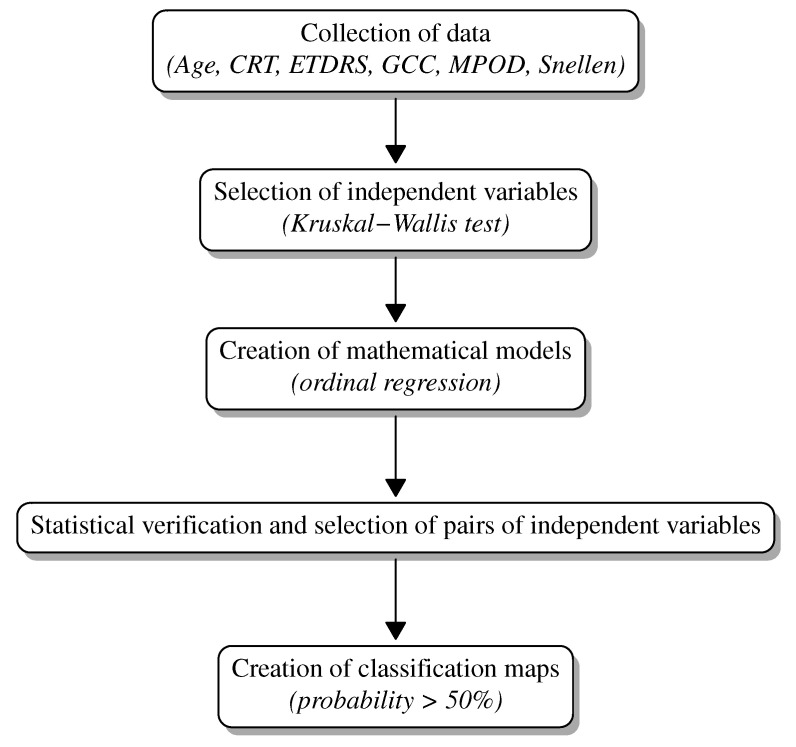
Flowchart of the method.

**Table 1 jpm-13-01074-t001:** Characteristics of patients (the median, minimum and maximum values of variables for each of the groups).

Variable	Control Group	AREDS 2	AREDS 3	AREDS 4
CRT (μm)	253.5 (214–306)	240 (194–286)	233 (131–307)	314.5 (93–660)
GCC (μm)	82.5 (74–88)	78 (63–92)	75 (62–88)	61.5 (22–89)
ETDRS	70 (50–85)	80 (35–85)	65 (35–85)	40 (5–76)
Snellen	0.50 (0.15–1)	0.8 (0.1–1)	0.4 (0.1–1)	0.125 (0.01–0.60)
MPOD (d.u.)	0.36 (0.10–0.62)	0.36 (0.0–0.7)	0.555 (0.0–0.9)	0.00 (0.00–0.72)
Age (years)	67.5 (61–85)	70 (62–92)	75 (61–84)	77 (59–87)

**Table 2 jpm-13-01074-t002:** Ordinal regression model coefficients for independent variables CRT and ETDRS.

Variable	Coefficient	*p*-Value	OR	2.5% CI	97.5% CI
CRT	0.006839	0.024910	1.006862	1.001133	1.013316
ETDRS	−0.066421	<0.000001	0.935736	0.912743	0.957144
Threshold coefficients					
Control group/AREDS 2		AREDS 2/AREDS 3		AREDS 3/AREDS 4	
−4.171311		−2.748292		−1.046439	

**Table 3 jpm-13-01074-t003:** Ordinal regression model coefficients for independent variables CRT and Snellen.

Variable	Coefficient	*p*-Value	OR	2.5% CI	97.5% CI
CRT	0.007693	0.006367	1.007722	1.002561	1.013819
Snellen	−2.676137	0.000007	0.068829	0.020959	0.216983
Threshold coefficients					
Control group/AREDS 2		AREDS 2/AREDS 3		AREDS 3/AREDS 4	
−0.884703		0.50022		2.035632	

**Table 4 jpm-13-01074-t004:** Ordinal regression model coefficients for independent variables CRT and Age.

Variable	Coefficient	*p*-Value	OR	2.5% CI	97.5% CI
CRT	0.011008	0.000301	1.011069	1.005563	1.017702
Age	0.080225	0.000337	1.083531	1.038016	1.133463
Threshold coefficients					
Control group/AREDS 2		AREDS 2/AREDS 3		AREDS 3/AREDS 4	
7.34911		8.713175		10.0887	

**Table 5 jpm-13-01074-t005:** Ordinal regression model coefficients for independent variables GCC and ETDRS.

Variable	Coefficient	*p*-Value	OR	2.5% CI	97.5% CI
GCC	−0.091017	0.012630	0.913002	0.846850	0.978064
ETDRS	−0.099617	0.000005	0.905184	0.863657	0.942014
Threshold coefficients					
Control group/AREDS 2		AREDS 2/AREDS 3		AREDS 3/AREDS 4	
−16.201571		−13.466137		−10.26158	

**Table 6 jpm-13-01074-t006:** Ordinal regression model coefficients for independent variables GCC and Snellen.

Variable	Coefficient	*p*-Value	OR	2.5% CI	97.5% CI
GCC	−0.081473	0.013013	0.921758	0.860098	0.976664
Snellen	−4.343067	0.000060	0.012997	0.001337	0.096795
Threshold coefficients					
Control group/AREDS 2		AREDS 2/AREDS 3		AREDS 3/AREDS 4	
−11.06145		−8.338816		−5.754173	

**Table 7 jpm-13-01074-t007:** Ordinal regression model coefficients for independent variables GCC and Age.

Variable	Coefficient	*p*-Value	OR	2.5% CI	97.5% CI
GCC	−0.065804	0.027764	0.936315	0.877389	0.987024
Age	0.118874	0.006711	1.126228	1.035529	1.231370
Threshold coefficients					
Control group/AREDS 2		AREDS 2/AREDS 3		AREDS 3/AREDS 4	
1.60897		3.884524		6.114169	

**Table 8 jpm-13-01074-t008:** Ordinal regression model coefficients for independent variables ETDRS and Age.

Variable	Coefficient	*p*-Value	OR	2.5% CI	97.5% CI
ETDRS	−0.069013	<0.000001	0.933315	0.910198	0.954920
Age	0.051857	0.021138	1.053225	1.008440	1.101663
Threshold coefficients					
Control group/AREDS 2		AREDS 2/AREDS 3		AREDS 3/AREDS 4	
−2.335994		−0.832369		0.804257	

**Table 9 jpm-13-01074-t009:** Ordinal regression model coefficients for independent variables Snellen and Age.

Variable	Coefficient	*p*-Value	OR	2.5% CI	97.5% CI
Snellen	−2.746244	0.000003	0.064168	0.019643	0.200547
Age	0.055591	0.013359	1.057166	1.012328	1.105757
Threshold coefficients					
Control group/AREDS 2		AREDS 2/AREDS 3		AREDS 3/AREDS 4	
1.150177		2.610493		4.065317	

## Data Availability

The data supporting the findings of the article are available upon request.

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
