# Peer review of "Classification Maps: A New Mathematical Tool Supporting the Diagnosis of Age-Related Macular Degeneration"

_jpm, 2023, doi:10.3390/jpm13071074_

Round 1

Reviewer 1 Report

The manuscript has a presented an approach to visualise trends associated with stages of AMD, developed using ordinal regression. I am unfortunately, struggling to appreciate the intent here. 

1. Given that the staging of AMD is reflected in multiple variables, what is the benefit of visualising two of them at a time? 

2. There has been a lot of effort in this space that looks that either using the entire OCT volume or OCTA volume to diagnose AMD. The additional of visual function is nice and I think necessary in models that stage AMD. Please emphasize how this approach is better or at least equivalent to existing methods. This paper presents a review of methods that rely on OCT: Srivastava, S., Divekar, A.V., Anilkumar, C. et al. Comparative analysis of deep learning image detection algorithms. J Big Data 8, 66 (2021). https://doi.org/10.1186/s40537-021-00434-w

3. I also am struggling with how this paper is written. The introduction is so heavily detailed on the pathogenesis of AMD and a history of ophthalmoscopy, but doesn't cover OCT and OCTA, which are the most widely used modalities clinical at the moment. Further, the description of the data is also extremely vague! How are the features that is utilised in the approach computed? Is the GCC is a total average or was it computed in sectors? What OCT system was utilised for this? How many patients did you have? Did any patients present with AMD in both eyes or at different stages? 

4. Given the multiple models developed and tested, why was no multiple testing correction used? 

Author Response

Comments and my answers (in bold) are attached (report_1_answers.pdf).

Reviewer 2 Report

This paper proposes a new mathematical tool based on the ordinal regression model for the diagnosis of age-related macular degeneration. Overall, the structure of this paper is clear. The main pipeline of this paper is easy to follow. However, there are still issues needed to be clarified:

1. It would be better, if the authors could add the main contributions of the paper at the end of the Introduction section.

2. The authors should discuss the current literature from 2021 to 2023. Please add more recent works to the Introduction section to enhance your literature review.

3. In the Materials and Methods section, the assumed significance level is 0.05. Why do you assume 0.05 for the significance level ? Other significance levels ? Please briefly explain its definition and role for the significance level.

4. I find that the right half of some of your figures was incomplete. Some of your figures need improvement, such as fig. 1, 2, 3, 7, and 8.

5. In the Results and Discussion section, I suggest you draw a flowchart for the proposed method. For a more detailed understanding of the proposed method, draw from the first step of receiving the input to the last step of the final output along with all the steps.

6. In the Conclusions section, the second paragraph: the quality of all classification models considered in this work is similar. The authors should declare the advantages and disadvantages of all classification models as the distinction is not clear.

7. Hyperspectral and multispectral systems also play an important role in classifification. For example, “Smartphone imaging spectrometer for egg/meat freshness monitoring” and “Open-source mobile multispectral imaging system and its applications in biological sample sensing”, it is suggested that hyperspectral and multispectral systems should be discussed.

8. The Conclusions section need polishing. Although proven to be working, the limitations and prospects should be described. Please add some limitations and prospects of your research work to the Conclusions section.

Author Response

Comments and my answers (in bold) are attached (report_2_answers.pdf).

Reviewer 3 Report

The paper has some technical specificity, but maybe is more adequate to a conference.

Introduction is not relevant to the variables used for the classification map:

1. CRT (Central Retinal Thickness),
2. GCC (Ganglion Cell Complex) tickness,
3. MPOD (Macular Pigment Optical Density),
4. ETDRS (Early Treatment Diabetic Retinopathy Study) chart
5. Snellen  chart
6. Age

I missed the verification of independence between the two variables used:

T2: CRT + ETDRS
T3: CRT + Snellen
T4: CRT + AGE

T5: GCC + ETDRS
T6: GCC + Snellen
T7: GCC + Age

T8: ETDRS + Age
T9: Snellen + Age

The conclusion of the paper is that only AREDS4 can be classified with high confidence.

Classification maps have some merits in the line of XAI.

I missed the connection between data in Tables and the explanation. For instance: ''. If the Age value increases by 1 unit, the odds

that the patient’s eye will be in a higher stage of AMD is about 8.4\% higher (Table 4).`` Clarify where 8.4 comes from the table.

''Quantities such as CRT ...`` - I assume only these quantifiers have been used.

The abbreviations are not uniform: Maybe GCCT, since CRT includes the term ''thickness`` in the abbreviation

Author Response

Comments and my answers (in bold) are attached (report_3_answers.pdf).

Round 2

Reviewer 1 Report

I appreciate the authors efforts to improve the introduction and description of the methods. 

However, I still do not understand why this 2D representation is needed to diagnose AMD. Wouldn't be easier to create a colour-coded grouping for all the variables used in AREDS, and just colour code the results for each patient? How is this different from the distribution of normals that is already available in a Cirrus report? 

Author Response

Thank you for the comment. A new text has been added to the 'Conclusions' section (marked in red in marked_version_2.pdf).

The definition of the distribution of normals in Cirrus report is given in the manual of Zeiss Cirrus HD-OCT:

"Distribution of Normals:
The thickest 5% of measurements fall in the white area.
90% of measurements fall in the green area.
The thinnest 5% of measurements fall in the yellow area or below.
The thinnest 1% of measurements fall in the red area. Measurements in red are considered outside normal limits.
ONH values will be shaded gray if disc area is not within the central 90% of normal range."

available at

https://www.zeiss.co.uk/content/dam/Meditec/gb/Chris/OCT%20Business%20Builder/PDF%27s/1.pdf

The probabilities presented by us are not related to this definition.  They are model-predicted probabilities of the occurrence of the stages of AMD.

If we were to create color-coded grouping for all variables used in AREDS and simply color-code the results for each patient, we would only get positions in the variable space used for description. In this way, we will not receive information on the dynamics of changes of the probability of belonging to AREDS stages as a dependence on the values of the variables used. In order to obtain such information, a mathematical model of changes of probabilities as a function of variables should be created, e.g. the one presented in our work based on ordinal regression. The tables of the presented work contain coefficients that describe this dynamics of changes. The presented maps result directly from the created models.

Analogously, the distribution of normals available in the Cirrus report does not predict changes of the AREDS stages as a function of the values of the variables used.